# Individual and intimate-partner factors associated with cervical cancer screening in Central Uganda

**Alone Isabirye***

Department of Sociology, Anthropology and Population Studies (Demography), Faculty of Social Sciences, Kyambogo University, Kampala, Uganda

* aloneisab@gmail.com

## Abstract

Intimate-partner factors have a significant effect on the uptake of services that affect maternal reproductive health outcomes. There is limited research on intimate-partner factors associated with cervical cancer screening. Therefore, this article examines the intimate-partner correlates of cervical cancer screening among married women in Central Uganda. We conducted a cross-sectional survey in Wakiso and Nakasongola districts in Central Uganda. A total of 656 married women aged 25–49 participated in the study. Frequency distributions for descriptive statistics and Pearson chi-squared tests were done to identify the association of selected individual explanatory variables and intimate-partner factors with cervical cancer screening. Finally, multivariable complementary log-log regressions were used to estimate intimate-partner factors associated with women's cervical cancer screening uptake in Central Uganda. About 2 in 10 (20%) of the participants had been screened for cervical cancer. The following characteristics when examined separately in relation to the uptake of cervical cancer screening service and were significant: woman's age, education attainment, occupation, wealth index, parity, male partner's age, and male partner's emotional support. After adjusting for independent factors, cervical cancer screening was significantly associated with women who had; attained secondary (AOR = 2.19; CI 1.18–4.06) compared to none/ primary education, and received partner's emotional support (AOR = 30.06; CI 13.44–67.20) compared to those who did not receive partner's emotional support. In Central Uganda, cervical cancer screening among married women was significantly associated with women's education, and partner's emotional support. These factors point to the importance of intimate-partner factors. Therefore, more effort should be directed at encouraging men's participation. This should be supplemented with empowering women through education to increase uptake of screening services.

## Background

Globally, cervical cancer is the fourth most common type of cancer with 569 847 new cases and 311 365 deaths annually among women. The global age-standardized incidence rate per

**Data Availability Statement:** All relevant data are within the paper and its Supporting Information files.

**Funding:** This work was funded by the Germany Academic Exchange Service (DAAD) (https://www.

daad.de) funding program in-country/ in-region scholarship program Uganda 2016 under grant number 57299305 and awarded to IA. The funders had no role in study design, data collection and analysis, decision to publish, or preparation of the manuscript.

**Competing interests:** The authors have declared that no competing interests exist.

100, 000 is 13.1 [1]. The Sub-Saharan African cervical cancer deaths constitutes 21.6% of the global cancer deaths; ranking the continent's burden in the second position after India (25.4%) [2]. Secondly, Sub-Saharan Africa has the region (Eastern Africa) with the biggest burden (52613 new cases) of the disease globally [2]. Despite the significant burden posed by cervical cancer, the international policy framing demonstrates the members' commitment to averting the disease [3, 4]. The international cervical cancer framing emphasizes the importance of cervical cancer screening for women age 25–49 years [4]. The significance of ensuring equity in access to sexual and reproductive health-care services was also fronted as a fundamental human right by the Sustainable Development Goals (SDGs), especially goals; 3 and 5 [3]. The international policy framing also recognized the importance of the intimate-partner in improving the maternal reproductive health outcome [5, 6]. The significance of intimate-partner factors in improving reproductive health first gained an international platform during the 1994 International Conference on Population and Development in Cairo [5]. Later, the 1995 women's Conference in Beijing agitated for shared responsibility between men and women on reproductive health matters to improve women's health [6]. The World Health Organization also recognized the importance of integrating men in the prevention of cervical cancer in Low and middle-income-economies [7].

Cervical cancer is the principal cause of cancer related deaths among women in most Low and Middle Income Countries (LMICs), Uganda inclusive [2]. Uganda is among the seven countries with the highest cervical cancer incidence rates [2]. Uganda's age-standardized incidence (54.8) and mortality (25) rate per 100, 000 is over four folds the global estimate of 13.1 and 6.8 respectively [1]. Close to half (40%) of all cancer cases recorded by Kampala Cancer Registry (KCR) are as a result of cervical malignancy [8]. The 2018 estimates indicated that about, 6413 new cases and 2400 deaths due to cervical cancer were registered in Uganda [9]. Additionally, about 80% of the female cancer patients at the Uganda Cancer Institute are diagnosed with cervical cancer. As Uganda aspires to attain the middle income status, the burden of cervical cancer is also likely to escalate with adoption of unhealthy lifestyles [10, 11]; Uganda's cervical cancer burden is projected to account for about, 6400 new cases and 4300 deaths annually by 2025 [12].

Widespread cytology-based screening has proven to be efficacious in reducing cervical cancer incidence and mortality in developed countries [13]. However, LMICs have not seen equivalent interventions, hence perpetuating inequality in the burden of cervical cancer between developed countries and LMICs [1]. The disparity in the burden of cervical cancer between developed countries and LMICs can be significantly narrowed if LMICs implement cervical cancer screening programs [13, 14]. Ensuring effectiveness of cervical cancer screening programs in LMCs will call for leveraging access and uptake [15, 16]. The Ministry of Health strategic plan for cervical cancer prevention and control 2010–2014 targeted to screen and treat 80% of eligible women aged 25–49 years by 2015 [17]. However, the uptake of services is still low (7%); according to study from Central Uganda [18]. The low uptake of cervical cancer screening is attributed to financial constraints, and lack of awareness related to cervical cancer screening [19]. Additionally, studies have indicated that inadequate male participation may be an underrated impediment to cervical cancer screening [20].

Several studies have underscored the role of intimate-partner factors in influencing reproductive health outcomes of women including; maternal and child health [21, 22], reproductive health [23, 24], labor preparation [25, 26], and contraceptive use [27]. The abovementioned studies point to scarcity of evidence regarding intimate-partner factors and cervical cancer screening. Research has associated cervical cancer screening with some intimate-partner factors; women who had experienced intimate partner violence were less likely to have ever screened compared to their counterparts in Kenya [37]. Some studies have also recognized the

intimate-partner factors associated with spouses' cervical cancer screening including; participating in decision-making, granting permission, providing transport, and financial support to attend health services cited by studies in; Ghana [28], Swaziland [29], Kenya [30], and among Sub-Saharan immigrant men in the United States of America (USA) [31]. Additionally, research has also associated cervical cancer screening with several individual factors. These include; women's age [32–34] and women's social economic status [32, 35]. Majority of the studies on cervical cancer screening uptake in Uganda focused solely on women without controlling for intimate-partner factors [36], vanishing the ground for testing the social support theory vis-à-vis cervical cancer screening.

Evidence has demonstrated that men influence the reproductive health outcomes of women; men impact women's reproductive health through their role as intimate-partners, fathers, and healthcare workers [26, 37–39]. This is because, in some developing countries with patriarchal structures, the effect of intimate-partner factors are even more pronounced. The husbands or other family members control women's health-related decisions, access to health care facilities and the use of family resources [39, 40]. The aforementioned literature on cervical cancer screening is inconclusive about the women and intimate-partners' factors of cervical cancer screening in Uganda. The main objective of the study was to establish women and intimate-partners' factors of cervical cancer screening in Central Uganda. The findings of the current study will inform effective interventions related to gender balance in men's and women's cervical cancer screening rights and responsibilities.

## Methods

We conducted a population-based cross sectional-survey in two of 27 districts in Central Uganda during June and July 2019. The two districts include Wakiso; a peri-urban area near Kampala city and Nakasongola district; a rural district. Cervical cancer prevention interventions have been implemented in these two districts [17]. Approval to conduct the study was obtained from Makerere University School of Social Sciences Research and Ethics Committee (MAHSSREC) and the Uganda National Council of Science and Technology (UNCST); UNCST registration number SS4848. Entry into the communities was cleared by the district as well as local authorities. Voluntary written informed consent was obtained from all participants, and they were assured of confidentiality before initiating interviews. Participants were also informed of their freedom to decline participation if they chose. These women were also abreast of their right to join the study and withdraw at any point without fear of retribution from the study team. The study participants were granted modest time (5–10 minutes) to ask questions about the study subject shortly after the interviews.

### Sample size and sampling procedure

The sample size of 850 women was calculated using Kish Leslie formula [41]. The sample size of 850 women was derived basing on 50% prevalence (P) of cervical cancer screening. The 50% prevalence was hypothesized because; no studies were found in literature for a similar population, and it provided the maximum possible sample size [42, 43]. Additionally, a precision of 5% was considered to allow for 95% confidence intervals around the estimates. Being a cluster random study, we also factored in the design effect of two and a response rate of 90% [42]. This response rate was considered adequate to allow for precision and generalizability of findings [44].

Basing on the principle of Probability Proportion Sampling (PPS) (mid-year population for women aged 25–49 and number of villages to Size) and the principle of optimum allocation

[45, 46] the overall sample size of 850 women (aged 25–49) was distributed on a ratio of 1 to 4 for Nakasongola district (250 women) and Wakiso district (600 women) respectively.

Out of the 850 women, data for only 656 married women were analyzed for the current study. The target population was women age 25–49 years. These were preferred because they are the high-risk-category to cervical cancer [47]. The 25–49 age group is also recommended by MOH for cervical cancer screening [17]. Computer-generated random number digits were used to randomly select 24 of the 1582 villages from Wakiso district and 10 of the 334 villages from Nakasongola district. Each selected village/ ward was considered a cluster and from each cluster, 25 households were selected using systematic random sampling.

## Data collection procedure

We collected data using a structured pre-tested questionnaire containing items adapted from findings and tools used in studies elsewhere [35, 36, 48, 49]. The Questionnaire was validated by 5 experts, who were familiar with the study subject. The experts reviewed the questionnaire and only questions that effectively captured issues related to cervical cancer prevention were maintained. Additionally, the questionnaire was pilot-tested with 10 women in the nearby setting. The questionnaire was pilot-tested in an area with characteristics similar to that of the study area. The pre-testing helped in fine-tuning the items of the questionnaire [50]. Cronbach's Alpha was also used to establish the internal consistency (reliability) of questions that were used to measure the construct (cervical cancer knowledge). A Cronbach's Alpha of 0.70 revealed that the items were good in measuring the construct [51].

The survey questionnaire comprised of seven sections. Section one had questions requiring respondents to provide their socio-demographic characteristics such as age, the highest level of education attained, marital status, type of residence, and history of health seeking. Questions on household characteristics such as type of house and household assets were presented in section two. Questions on reproduction such as the number of children, and contraception use were laid in section three. Questions on; cervical cancer knowledge and awareness, cervical cancer screening, and HPV vaccination were presented in the fourth, fifth, and sixth sections respectively. Finally, questions in section seven captured the husbands' characteristics.

The questionnaire was designed in English yet, the interviews were conducted in Luganda-the main local language spoken in Central Uganda. Therefore, the English version of the questionnaire was translated into Luganda language by two natives (well versed with Luganda). The questionnaire was back-translated into English by a pair of translators not exposed to the original version. The two versions of the questionnaire were compared for conceptual equivalence and differences were harmonized. A final version of the translated questionnaire was checked for accuracy and preservation of meanings.

Five female Research assistants (RAs) with Bachelor degrees in social sciences and education were recruited. The RAs were trained for 2 days on principles of quantitative and survey research. Research assistants were also abreast on the objectives of the research, sampling procedures, interview techniques and consent procedures. The RAs were deployed in the pre-selected study villages and each RA collected data from 6 to 8 participants per day, for a period of 28 days. To ensure quality and comparability of data between the RAs, the Principle Investigator (PI) reviewed the collected data on a daily basis.

The outcome variable of the study was cervical cancer screening. Cervical cancer screening was measured in terms of whether respondents underwent any cervical cancer examination ever; respondents were specifically asked "Have you ever been tested or examined for cervical cancer?"(No/Yes). Explanatory variables included; women's age (≤29, 30–39, and 40–49 years), intimate-partners' age (≤34, 35–49 and ≥50), religion (Roman Catholicism,

Pentecostals, Islam, Protestant, Islam, and others), study site/ type of residence (Wakiso/ urban and Nakasongola/ rural), education level (none/ primary, secondary, and post-secondary), ethnicity (Baganda, Banyankole, Basoga, Baluri and others), age at first marriage (≤18, 19–34 and singles) number of living children (0–3 and ≥4). Responses on use of family planning, visiting a health facility in the last six months, and having co-wives were got by asking respondents questions that required No/ Yes responses. Regarding decision making dependence, women were asked on who decides on their own health (1 = woman decides alone/ jointly with partner, 0 = partner alone/ others). Women's dependence in decision-making was considered low if the response was individual or joint participation, and high to those who responded partner/ others. Regarding intimate-partners' factors, intimate-partner support was measured in terms of whether respondents had ever received support for cervical cancer screening from their intimate-partners? (No/Yes). Those who answered yes were asked the type of support (financial, encouragement/ emotional or others) they had received from their intimate-partners. Wealth index was a composite score measured by household assets such as televisions, bicycles, materials used for housing construction and other characteristics related to wealth. Factor scores of household assets were generated. For the current study, it was recoded into three tertiles: poor, middle and rich.

## Data management and analysis

Two independent clerks entered data using Epidata 3.1 (EpiData Software, Odense, Denmark).

Data were synchronized and cleaned and then exported to STATA I/C version 16 for analysis [52]. Descriptive statistics in form of frequencies were generated. Chi-squared tests were then used to determine associations between independent variables and dependent variable; cervical cancer screening. Variables that were significant at 0.05% were considered for the multi-variable complementary log-log regression model to determine the magnitudes of associations between independent and dependent variables; odds ratios were reported with accompanying 95% confidence intervals. The complementary log-log regression model deemed suitable while dealing with rare outcomes or, when the data are asymmetrical, and the outcome is binary [53]. In this case, cervical cancer screening among married women had an uneven distribution of 20%.

## Results

### Distribution of study participants by demographic, socio-economic, and intimate-partner factors

Results in Table 1 shows that, majority of the respondents were from an urban setting (Wakiso district) (70%), in the middle wealth class (63.4%), and had ≤3 children (69%). Most women used contraceptives (60.5%) and their age at first marriage was 19–34 years (64.5%). Half of the women had attained secondary education (50%), approximately 3 in 10 (30.6%) were Protestants, 41.8% were engaged in business, 37% were Baganda and 45.9% were aged 30–39. Four in ten (40.4%) of the intimate-partners were engaged in business. The majority of the intimate-partners were also aged 35–49 (63%), had attained secondary education (54.7%), had one wife (74.7%), were the decision makers regarding spouses' health (79.3%), and had never offered screening emotional support (97.7).

Table 1 also shows the results of the cross tabulation (chi-squared tests) between demographic factors, socio-economic factors, intimate-partner factors and cervical cancer screening. Cervical cancer screening was significantly associated with women's age (p = 0.014),

**Table 1. Distribution of study participants by demographics, socio-economic factors, intimate-partner factors and cervical cancer screening status (N = 656).**

| Characteristics | % of women | Frequency | % Screened | P-value |
|---|---|---|---|---|
| **Age group** | | | | **0.014** |
| ≤29 | 38.1 | 250 | 14.4 | |
| 30–39 | 45.9 | 301 | 22.9 | |
| 40–49 | 16.0 | 105 | 25.7 | |
| **Religion** | | | | **0.881** |
| Catholics | 24.5 | 161 | 21.7 | |
| Protestants | 30.6 | 201 | 18.9 | |
| Muslims | 20.4 | 134 | 17.9 | |
| Pentecostals | 20.7 | 136 | 22.1 | |
| Others | 3.7 | 24 | 20.1 | |
| **Study site** | | | | **0.892** |
| Wakiso | 70.0 | 459 | 20.3 | |
| Nakasongola | 30.0 | 197 | 19.8 | |
| **Education attainment** | | | | **0.009** |
| None/ primary | 40.4 | 265 | 18.9 | |
| Secondary | 50.0 | 328 | 18.3 | |
| Post-secondary | 9.6 | 63 | 34.9 | |
| **Ethnicity** | | | | **0.167** |
| Baganda | 37.0 | 243 | 20.2 | |
| Banankole | 14.3 | 94 | 19.2 | |
| Basoga | 9.5 | 62 | 30.7 | |
| Baluri | 22.4 | 147 | 20.4 | |
| Others | 16.8 | 110 | 14.6 | |
| **Age at first marriage** | | | | **0.857** |
| ≤18 | 35.5 | 233 | 19.7 | |
| 19–34 | 64.5 | 423 | 20.3 | |
| **Wealth index** | | | | **0.010** |
| Poor | 20.7 | 136 | 16.2 | |
| Middle | 63.4 | 416 | 18.8 | |
| Rich | 15.9 | 104 | 30.8 | |
| **Currently using contraception** | | | | **0.273** |
| No | 39.2 | 257 | 17.9 | |
| Yes | 60.5 | 397 | 21.4 | |
| **Parity** | | | | **0.075** |
| ≤3 | 56.9 | 373 | 17.7 | |
| ≥4 | 43.1 | 283 | 23.3 | |
| **Intimate-partner's age** | | | | **0.009** |
| ≤34 | 35.8 | 235 | 13.3 | |
| 35–49 | 63.0 | 413 | 23.8 | |
| ≥50 | 11.3 | 74 | 21.3 | |
| **Intimate-partner's education** | | | | **0.034** |
| Primary | 31.9 | 209 | 21.1 | |
| Secondary | 54.7 | 359 | 17.3 | |
| Above secondary | 13.4 | 88 | 29.6 | |
| **Intimate-partner's occupation** | | | | **0.130** |
| Not employed | 2.9 | 19 | 26.3 | |
| Farmers | 19.5 | 128 | 26.6 | |

(*Continued*)

**Table 1.** (Continued)

| Characteristics | % of women | Frequency | % Screened | P-value |
|---|---|---|---|---|
| Professionals | 7.9 | 52 | 25.0 | |
| Business | 40.4 | 265 | 16.2 | |
| Other occupation | 29.3 | 192 | 19.3 | |
| **Intimate-partner has many wives** | | | | **0.720** |
| No | 74.7 | 490 | 19.8 | |
| Yes | 25.3 | 166 | 21.1 | |
| **Male intimate-Partner decides on spouse's own health** | | | | **0.570** |
| No | 20.7 | 136 | 18.4 | |
| Yes | 79.3 | 520 | 20.6 | |
| **Intimate-Partner ever encouraged spouse to go for screening** | | | | **0.000** |
| No | 97.7 | 641 | 18.7 | |
| Yes | 2.3 | 15 | 80.0 | |

education attainment (p = 0.009), occupation (p = 0.018), wealth index (p = 0.010) and parity (0.075). Intimate-Partner's factors that were significantly associated with spouse's cervical cancer screening included age (p = 0.009), education attainment (p = 0.034), and obtaining screening related emotional support (p<0.001).

Cervical cancer screening was higher among women aged 40–49 (25.7%), who had attained post-secondary education (29.6%), the rich (30.8%) and women with 4 or more children (23.3%). Cervical cancer screening was also higher among women whose spouses were age 35–49 years (23.8%), and had attained post-secondary education (34.9%). Cervical cancer screening was also higher among women whose spouses had extended screening related emotional support (80%) and screening related financial support (94.3%). Women's religion, study site, ethnicity, age at first marriage and use of contraception were not significantly associated with cervical cancer screening. Intimate-partner's decision-making authority, number of wives and occupation were also not significantly associated with cervical cancer screening.

## Associations between socio-demographic, economic, and male partner factors with cervical cancer screening

In Table 2, we estimated complementary log-log regression models to examine the relationship between cervical cancer screening and women's background characteristics, controlling for male partner factors. The first model consisted of women's background characteristics namely; age, education attainment, wealth index, parity and occupation. In the final model, we added male partner factors. None of women's characteristics consistently retained its significance after controlling for male partners characteristics. Women's education lost significance after controlling for male partners characteristics. In the first model, women who had attained post-secondary education had increased odds of having ever screened compared with women who had attained primary education (OR = 2.00; CI 1.07–3.76).

In the final model, women's education remained significant after controlling for male partner characteristics. Women who had attained secondary education had increased odds of having screened compared with women who had attained primary education (OR = 2.19; CI 1.18–4.06). Women whose partners had extended screening related emotional support had increased odds of having screened compared with their counterparts who did not receive spouse emotional support (OR = 30.06; CI 13.44–67.20). Women whose partners had extended

**Table 2. Associations between socio-demographic, economic and male partner factors with cervical cancer screening.**

| Characteristic | Model 1 | | Model 2 | |
|---|---|---|---|---|
| | OR | 95% CI | OR | 95% CI |
| **District** | | | | |
| Nakasongola (Ref) | | | | |
| Wakiso | 1.19 | 0.65–2.19 | 1.32 | 0.73–2.38 |
| **Age group** | | | | |
| ≤ 29 (Ref) | | | | |
| 30–39 | 1.55 | 1.00–2.41 | 1.16 | 0.60–2.25 |
| 40–49 | 1.81 | 0.99–3.32 | 1.46 | 0.58–3.65 |
| **Education attainment** | | | | |
| None/ Primary (Ref) | | | | |
| Secondary | 1.16 | 0.75–1.78 | 2.19* | 1.18–4.06 |
| Post-secondary | 2.00* | 1.07–3.76 | 2.52 | 0.98–6.49 |
| **Wealth index** | | | | |
| Poor (Ref) | | | | |
| Middle | 1.11 | 0.67–1.81 | 0.72 | 0.38–1.36 |
| Rich | 1.65 | 0.93–2.93 | 0.68 | 0.32–1.47 |
| **Parity** | | | | |
| ≤3 (Ref) | | | | |
| ≥4 | 1.18 | 0.77–1.81 | 1.34 | 0.75–2.39 |
| **Intimate-partner's age group** | | | | |
| ≤ 34 (Ref) | | | | |
| 35–49 | | | 1.53 | 0.77–3.04 |
| ≥50 | | | 1.66 | 0.59–4.69 |
| **Intimate-partner education attainment** | | | | |
| None/ Primary (Ref) | | | | |
| Secondary | | | 1.02 | 0.58–1.77 |
| Post-secondary | | | 0.98 | 0.58–1.65 |
| **Intimate-partner ever encouraged spouse to go for screening** | | | | |
| No (Ref) | | | | |
| Yes | | | 30.06*** | 13.44–67.20 |
| **Intimate-partner ever given financial support for screening** | | | | |
| No (Ref) | | | | |
| Yes | | | 53.94*** | 30.66–94.87 |

*p < 0.05

**p < 0.01

***p < 0.001.

RC = reference category

OR = Odds Ratios

CI = confidence interval

screening related financial support had increased odds of having screened compared with their counterparts (OR = 53.94; CI 30.66–94.87). Women's age, wealth index, occupation and parity were not significantly associated with cervical cancer screening in any of the two models. The partners' age and education attainment were also not significantly associated with cervical cancer screening.

## Discussion

The present study sought to establish the intimate-partner factors associated with cervical cancer screening among married women in Central Uganda. Women's education attainment, intimate-partners' financial and emotional support were significantly associated with cervical cancer screening among married women in Central Uganda. Our findings indicate that only 20% of the married women in Central Uganda had ever screened for cervical cancer. This finding from the current study is higher than findings from Eastern Uganda (4.8%) [36], and Zimbabwe (9%) [35]. Most of these studies were health facility based, focus on one type of residence, and had small sample sizes. However, our findings are close to findings published from Tanzania (22.6%) (20), Kenya (19.4%) [32] and Nepal (18.3%) [33]. The low cervical cancer screening rate among married women in Central Uganda is of great concern and attests to the need to scale-up cervical cancer screening services to all health facilities in Central Uganda and other regions of the country.

When woman and intimate-partner's factors were examined simultaneously in relation to cervical cancer screening, only women's education attainment and intimate-partners' emotional and financial support were significantly associated with screening status. It was found that the likelihood of screening increased among women with secondary, and post-secondary compared with their counterparts with none/ primary education. The possible explanation is that, educated women are more likely to be empowered and informed about their health [49, 54–56]. Our finding is consistent with findings from other studies [33, 57]. However, our study results are not supported by findings from El Salvador study [58]. The difference in the findings may be explained by the context in which the two studies were done; the El-Salvador study was done in an area with cervical cancer screening program, yet the current study was done in an area without organized cervical cancer screening program. Our result point to the need for awareness campaigns in communities targeting illiterate women or those with primary education to motivate them to embrace cervical cancer screening.

In this study, two different dimensions of spousal/ social support were significantly associated with cervical cancer screening. First, the likelihood of cervical cancer screening increased among women who had benefited from intimate-partners' social support in the form of information compared with their counterparts. Our research concurs with findings from low-resource countries (synthesis of research projects in Bolivia, Peru, Kenya, South Africa, and Mexico) [59]. Our finding is also supported by results from studies done in Nigeria, which indicated that women required spousal-information-support before completing cervical cancer screening [60, 61]. Our research underscores the spousal/ social support function of social networks, especially marital relationships [62]. Therefore, our study point to the importance of making social networks an integral component in the interventions that aim at increasing cervical cancer screening. Additionally, the likelihood of cervical cancer screening was higher among women who reported receiving spousal financial support compared with their counterparts. Financial support is classified by the social support theory as instrumental support and is believed to have a positive influence on health [62]. Our result is supported by previous studies where women cited financial constraints as barriers to cervical cancer screening [19, 29, 31, 63, 64]. Our study findings suggest that awareness and health education programs on women's health needs should consider and involve men; to reinforce intimate-partners' roles in cervical cancer screening. Our study examined intimate-partner's cervical cancer screening support, yet social support research through the relationship perspective has suggested that health outcomes of social support are not independent of relationship processes; they simultaneously occur with support, such as intimacy, companion, and low social conflict, calling for further research in this area [62]. Contrary to our hypothesis, intimate-partners' education attainment

was not significantly associated to their wives' cervical cancer screening, though the odds ratios indicated that the likelihood of screening was higher among women whose intimate-partners had secondary education. This may point to the minimum level of intimate-partner's formal education necessary to positively influence the health outcomes of their spouses. Our study results are in consonance with previous studies [29] which did not find significant relationship of intimate-partners' age with their spouses' screening status. However, our study findings are not supported by findings from Tanzania [24]; this study found a positive, significant relationship between cervical cancer screening and intimate-partners' age.

Surprisingly, wealth status was not significantly associated with cervical cancer screening, though the odds ratios indicated a positive relationship between wealth index and cervical cancer screening. This is because higher social economic status is associated with access to health services. A positive relationship between wealth status and cervical cancer screening was found in Eastern Jamaica [34], Zimbabwe [35], and Kenya [32], though statistically significant. It was hypothesized that higher parity was significantly associated with cervical cancer screening. Surprisingly, the study results show that parity was not significantly associated with cervical cancer screening though, the odds ratios indicated a higher likelihood of cervical cancer screening among women of higher parity. Studies from elsewhere indicated a positive, significant relationship between parity and cervical cancer screening [33, 34]. Higher parity coincides with older women who are most likely to see health workers after experiencing signs and symptoms [33]. However, our findings are inconsistent with findings from Jamaica [34] and Nepal [33] which found a positive significant relationship between parity and cervical cancer screening. Type of residence was not significantly associated with cervical cancer screening though, the odds of cervical cancer screening were higher among urban dwelling women. Our findings are not supported by findings from Tanzania [21], where maternity care was significantly associated with type of residence. Age was not significantly associated with cervical cancer screening, though the odds for cervical cancer screening were higher among older women compared to their younger counterparts. This finding indicate that screening uptake is independent of age; other factors may be responsible for screening among both the old and young women [65]. However, this finding is contrary to studies from elsewhere [32, 34] as they indicate a significant influence of age on cervical cancer screening.

## Study strengths and limitations

The current study provided evidence on the individual as well as the intimate-partner factors as correlates of cervical cancer screening. The distribution of the outcome variable was rare (20%). Therefore, the use of the log-log regression penalized the rare nature of the outcome variable and provided more precise estimates. Despite the strengths of the study, there were limitations worthy noting. First, this study is cross-sectional, therefore, it is not possible to assess causality. Secondly, although the study had a large sample size (N = 656), there was high probability for cervical cancer screening when it came to intimate-partner support factors, which affected the statistical tests and led to some wide confidence intervals on two variables. Finally, this study was carried out in two majorly two districts (Urban and rural) in central Uganda and therefore, the study findings may not be generalized to other contextually different areas.

## Conclusion

This study found low uptake of cervical cancer screening (20%). In Central Uganda, cervical cancer screening among married women was significantly associated with women and intimate-partner factors; women's education attainment, intimate-partner's emotional and

financial support were significantly associated with cervical cancer screening. The intimate-partner's factors associated with cervical cancer screening point to traditional marital roles of male partner dominance. Efforts to encourage men's participation through community education is recommended. This should be supplemented with empowering women through education to increase uptake of screening services.

## Supporting information

**S1 File. English questionnaire.**
(DOCX)

**S2 File. Translated questionnaire.**
(DOCX)

**S1 Dataset.**
(XLS)

## Author Contributions

**Conceptualization:** Alone Isabirye.

**Data curation:** Alone Isabirye.

**Formal analysis:** Alone Isabirye.

**Funding acquisition:** Alone Isabirye.

**Investigation:** Alone Isabirye.

**Methodology:** Alone Isabirye.

**Project administration:** Alone Isabirye.

**Resources:** Alone Isabirye.

**Software:** Alone Isabirye.

**Supervision:** Alone Isabirye.

**Validation:** Alone Isabirye.

**Visualization:** Alone Isabirye.

**Writing – original draft:** Alone Isabirye.

**Writing – review & editing:** Alone Isabirye.

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
