## [Decision Letter · Decision Letter 0]

19 May 2022

PONE-D-21-33110

The role of the male partner and cervical cancer screening among married women in Central Uganda

PLOS ONE

Dear Dr. Isabirye,

Thank you for submitting your manuscript to PLOS ONE. After careful consideration, we feel that it has merit but does not fully meet PLOS ONE’s publication criteria as it currently stands. Therefore, we invite you to submit a revised version of the manuscript that addresses the points raised during the review process.

Please carefully revise the paper according to reviewers' suggestions and resubmit within 30 days.

The title of the study should be in line with the methodology, findings and conclusion. Also, the expression of English language be revised throughout the manuscript for its better understanding. 

We look forward to receiving your revised manuscript.

Kind regards,

Muhammad Farooq Umer,

BDS, MSPH, FRSPH, PhD Epidemiology and Health Stat

Guest Editor

PLOS ONE

**Journal requirements:**

**Reviewers' comments:**

Reviewer's Responses to Questions

**Comments to the Author**

1. Is the manuscript technically sound, and do the data support the conclusions?

Reviewer #1: Yes

Reviewer #2: No

2. Has the statistical analysis been performed appropriately and rigorously? 

Reviewer #1: Yes

Reviewer #2: I Don't Know

3. Have the authors made all data underlying the findings in their manuscript fully available?

Reviewer #1: Yes

Reviewer #2: No

4. Is the manuscript presented in an intelligible fashion and written in standard English?

Reviewer #1: Yes

Reviewer #2: No

5. Review Comments to the Author

Reviewer #1: This study examined the role of men in cervical cancer screening among married women in Central Uganda using cross-sectional survey. The comments are as follows.

Background

Why should the authors conduct the research? The knowledge gap between previous studies and study aim should be mentioned.

The study aim (purpose) should be clearly mentioned in the end of background.

Study design

In calculating sample size, the authors stated that “The prevalence of cervical cancer screening was estimated at 50% because no studies were found in literature for similar population.” Why did the authors use 50%? Even if the authors have such claim above, they also need to mention the source of “50%”, which may from other population. The number of participants (study subjects) should be higher (usually 10%-20%) than the calculated sample size for possible incomplete data collection and withdrawn of study subjects. Have the authors considered it?

Have the authors referred to other well-developed questionnaire in designing the questionnaire in the study? If yes, the reference should be noted.

Results

In the tables, please use three-line table.

Discussion

What are the strengths of the study? This should be mentioned with limitations of the study.

Others

The manuscript need proofreading before submission.

The format of the reference should be carefully checked one by one, the mistakes and format issue should be revised.

Reviewer #2: The paper's title is inconsistent with the study findings. The author studied factors associated with cervical cancer screening among women in the two study districts of which male partner's age was one of the significant factors but opted to present the factor in the context of male involvement. Besides, if the study was on male involvement, why did the author study women?

6. PLOS authors have the option to publish the peer review history of their article (what does this mean?). If published, this will include your full peer review and any attached files.

Reviewer #1: No

Reviewer #2: No

---

## [Author Response · Author response to Decision Letter 0]

18 Jun 2022

Response letter

Date 18 June 2022

To: PLOS ONE em@editorialmanager.com

From: "Alone Isabirye" aloneisab@gmail.com

Subject: Response to review comments of our manuscript submitted to PLOS ONE (PONE-D-20-2133110)

Male partner correlates of cervical cancer screening in Central Uganda

Alone Isabirye, 

Dear Editor,

Thank you for your reply regarding my manuscript "Male partner correlates of cervical cancer screening in Central Uganda". I am grateful for your and the reviewers’ comments. I have revised and modified the manuscripts according to your and reviewer’s comments. As a consequence, I provide a revised manuscript with your and the reviewers’ suggestions integrated therein:

Response to editor’s comments

The title of the study should be in line with the methodology, findings and conclusion. Also, the expression of English language be revised throughout the manuscript for its better understanding. 

The title was revised to reflect the methodology, findings and conclusions.

The manuscript was written following the PLOSONE style.

Information about funding was systematized.

The ethics statement is presented in the methods section.

Captions for supporting information files were included at the end of the manuscript.

Response to reviewer 1’s comments

Background

Why should the authors conduct the research? The knowledge gap between previous studies and study aim should be mentioned.

The knowledge gap between previous studies and study aim was mentioned.

The study aim (purpose) should be clearly mentioned in the end of background.

The main objective of the study was clearly stated.

Study design

In calculating sample size, the authors stated that “The prevalence of cervical cancer screening was estimated at 50% because no studies were found in literature for similar population.” Why did the authors use 50%? Even if the authors have such claim above, they also need to mention the source of “50%”, which may from other population. The number of participants (study subjects) should be higher (usually 10%-20%) than the calculated sample size for possible incomplete data collection and withdrawn of study subjects. Have the authors considered it?

The 50% prevalence rate was justified with relevant literature.

The sample size formula I deployed in calculating the sample size catered for the non-response rate. The non-response rate was indeed fixed at 10%; in other words, a response rate of 90% was considered.

Have the authors referred to other well-developed questionnaire in designing the questionnaire in the study? If yes, the reference should be noted.

The questionnaire contains items used by previous studies, and this was supported with relevant literature.

Results

In the tables, please use three-line table.

All results-tables were presented as three-line tables.

Discussion

What are the strengths of the study? This should be mentioned with limitations of the study.

The strengths of the study were presented.

Others

The manuscript need proofreading before submission.

The manuscript was proofread and all omissions and grammatical issues were fixed.

The format of the reference should be carefully checked one by one, the mistakes and format issue should be revised.

All references were checked and perfected.

Response to reviewer 2’s comments

The paper's title is inconsistent with the study findings. The author studied factors associated with cervical cancer screening among women in the two study districts of which male partner's age was one of the significant factors but opted to present the factor in the context of male involvement. Besides, if the study was on male involvement, why did the author study women?

The title was revised to reflect the study findings.

Women were interviewed instead of men because they provided data related to their partners without necessarily contacting men. 

I hope that my modifications render my manuscript in its current form suitable for publication in PLOSONE

Yours sincerely,

Isabirye Alone

aloneisab@gmail.com

---

## [Decision Letter · Decision Letter 1]

12 Jul 2022

PONE-D-21-33110R1Male partner correlates of cervical cancer screening among married women in Central UgandaPLOS ONE

Dear Dr. %Isabirye%,

Thank you for submitting your manuscript to PLOS ONE. After careful consideration, we feel that it has merit but does not fully meet PLOS ONE’s publication criteria as it currently stands. Therefore, we invite you to submit a revised version of the manuscript that addresses the points raised during the review process.

ACADEMIC EDITOR:The previous Reviewer's comments (Reviewer 2) were not satisfactorily addressed. Another reviewer was invited and the new comments (Reviewer 3) are also not very encouraging. I urge you to carefully address all the concerns by responding to each query of previous as well as new reviewer. The publication at this stage depends upon the quality of response from you.All the comments are clear and self-explanatory, but if you need further clarity, please feel free to contact us.==============================

We look forward to receiving your revised manuscript.

Kind regards,

Muhammad Farooq Umer

PhD Epidemiology and Health Statistics

Academic Editor

PLOS ONE

Reviewers' comments:

Reviewer's Responses to Questions

**Comments to the Author**

1. If the authors have adequately addressed your comments raised in a previous round of review and you feel that this manuscript is now acceptable for publication, you may indicate that here to bypass the “Comments to the Author” section, enter your conflict of interest statement in the “Confidential to Editor” section, and submit your "Accept" recommendation.

Reviewer #1: All comments have been addressed

Reviewer #3: (No Response)

2. Is the manuscript technically sound, and do the data support the conclusions?

Reviewer #1: Yes

Reviewer #3: Yes

3. Has the statistical analysis been performed appropriately and rigorously? 

Reviewer #1: Yes

Reviewer #3: Yes

4. Have the authors made all data underlying the findings in their manuscript fully available?

Reviewer #1: Yes

Reviewer #3: Yes

5. Is the manuscript presented in an intelligible fashion and written in standard English?

Reviewer #1: Yes

Reviewer #3: No

6. Review Comments to the Author

Reviewer #1: The authors have addressed all the comments of the reviewer and the manuscript can be accepted after proof reading.

Reviewer #3: Comments

This research article signifies the women and the partner’s factors associated with the cervical HPV screening in the underprivileged population of Central Uganda having high burden of the cancer. The article has already been through the major revision and the author though has claimed to have addressed the major observations marked previously. The study is reasonably constructed and can be published but the author has to address the major observations noted herein again as follow.

Title

Title of the article has not been revised as suggested by both the previous reviewers. Streamline the literature presented with suitable title.

The red highlights in the “Revised Manuscript with Track Changes“ in the pdf file mostly are the same as presented in the original submission.

Improve the expression of English suggested by the Reviewer 1 as the red highlights are not changes but are consistent with the original submission.

Abstract:

P2L3 Replace capital M in male with small letter. Revise English language for better understanding.

Background

Track changes appearing red are the same as the original submission.

Highlight the knowledge gap and rationalize your study as previously suggested

Methods

Cite Reierson Draugalis et al. in the references in the sub heading of “Sample size and sampling procedure.

The author adopted the questionnaire from the previous studies. Was the translated questionnaire checked for reliability and validity? And what about the internal consistency, pilot test and Cronback’s alpha?

Results

Tables are not presented as three-line tables.

Discussion

Adequate

Conclusion

Adequate

References

Format of references not rectified and missing journal name, volume, issue and page number should be mentioned (Ref 1, Ref 14, Ref 21, Ref 28, Ref 34, Ref 38, Ref 49, Ref 43, Ref 46, Ref 50, Ref 56, Ref 58 and others

7. PLOS authors have the option to publish the peer review history of their article (what does this mean?). If published, this will include your full peer review and any attached files.

Reviewer #1: No

Reviewer #3: No

---

## [Author Response · Author response to Decision Letter 1]

20 Aug 2022

Response letter

Date 18 June 2022

To: PLOS ONE em@editorialmanager.com

From: "Alone Isabirye" aloneisab@gmail.com

Subject: Response to review comments of our manuscript submitted to PLOS ONE (PONE-D-20-2133110)

Male partner correlates of cervical cancer screening in Central Uganda

Alone Isabirye, 

Dear Editor,

Thank you for your reply regarding my manuscript "Male partner correlates of cervical cancer screening in Central Uganda". I am grateful for your and the reviewers’ comments. I have revised and modified the manuscripts according to your and reviewer’s comments. As a consequence, I provide a revised manuscript with your and the reviewers’ suggestions integrated therein:

Response to editor’s comments

The title of the study should be in line with the methodology, findings and conclusion. Also, the expression of English language be revised throughout the manuscript for its better understanding. 

The title was revised to reflect the methodology, findings and conclusions.

The manuscript was written following the PLOSONE style.

Information about funding was systematized.

The ethics statement is presented in the methods section.

Captions for supporting information files were included at the end of the manuscript.

Response to reviewer 1’s comments

Background

Why should the authors conduct the research? The knowledge gap between previous studies and study aim should be mentioned.

The knowledge gap between previous studies and study aim was mentioned.

The study aim (purpose) should be clearly mentioned in the end of background.

The main objective of the study was clearly stated.

Study design

In calculating sample size, the authors stated that “The prevalence of cervical cancer screening was estimated at 50% because no studies were found in literature for similar population.” Why did the authors use 50%? Even if the authors have such claim above, they also need to mention the source of “50%”, which may from other population. The number of participants (study subjects) should be higher (usually 10%-20%) than the calculated sample size for possible incomplete data collection and withdrawn of study subjects. Have the authors considered it?

The 50% prevalence rate was justified with relevant literature.

The sample size formula I deployed in calculating the sample size catered for the non-response rate. The non-response rate was indeed fixed at 10%; in other words, a response rate of 90% was considered.

Have the authors referred to other well-developed questionnaire in designing the questionnaire in the study? If yes, the reference should be noted.

The questionnaire contains items used by previous studies, and this was supported with relevant literature.

Results

In the tables, please use three-line table.

All results-tables were presented as three-line tables.

Discussion

What are the strengths of the study? This should be mentioned with limitations of the study.

The strengths of the study were presented.

Others

The manuscript need proofreading before submission.

The manuscript was proofread and all omissions and grammatical issues were fixed.

The format of the reference should be carefully checked one by one, the mistakes and format issue should be revised.

All references were checked and perfected.

Response to reviewer 2’s comments

The paper's title is inconsistent with the study findings. The author studied factors associated with cervical cancer screening among women in the two study districts of which male partner's age was one of the significant factors but opted to present the factor in the context of male involvement. Besides, if the study was on male involvement, why did the author study women?

The title was revised to reflect the study findings.

Women were interviewed instead of men because they provided data related to their partners without necessarily contacting men. 

I hope that my modifications render my manuscript in its current form suitable for publication in PLOSONE

Yours sincerely,

Isabirye Alone

aloneisab@gmail.com

---

## [Decision Letter · Decision Letter 2]

1 Sep 2022

Individual and intimate-partner factors associated with cervical cancer screening in Central Uganda

PONE-D-21-33110R2

Dear Dr. Isabirye,

We’re pleased to inform you that your manuscript has been judged scientifically suitable for publication and will be formally accepted for publication once it meets all outstanding technical requirements.

Kind regards,

Muhammad Farooq Umer, BDS, MSPH, FRSPH, PhD Epidemiology and Health Stat

Academic Editor

PLOS ONE

Additional Editor Comments (optional):

Reviewers' comments:

Reviewer's Responses to Questions

**Comments to the Author**

1. If the authors have adequately addressed your comments raised in a previous round of review and you feel that this manuscript is now acceptable for publication, you may indicate that here to bypass the “Comments to the Author” section, enter your conflict of interest statement in the “Confidential to Editor” section, and submit your "Accept" recommendation.

Reviewer #3: All comments have been addressed

2. Is the manuscript technically sound, and do the data support the conclusions?

Reviewer #3: Yes

3. Has the statistical analysis been performed appropriately and rigorously? 

Reviewer #3: Yes

4. Have the authors made all data underlying the findings in their manuscript fully available?

Reviewer #3: Yes

5. Is the manuscript presented in an intelligible fashion and written in standard English?

Reviewer #3: Yes

6. Review Comments to the Author

Reviewer #3: (No Response)

7. PLOS authors have the option to publish the peer review history of their article (what does this mean?). If published, this will include your full peer review and any attached files.

Reviewer #3: No

---

## [Editor Report · Acceptance letter]

7 Sep 2022

PONE-D-21-33110R2 

Individual and intimate-partner factors associated with cervical cancer screening in Central Uganda 

Dear Dr. Isabirye:

I'm pleased to inform you that your manuscript has been deemed suitable for publication in PLOS ONE. Congratulations! Your manuscript is now with our production department. 

Kind regards, 

on behalf of

Dr. Muhammad Farooq Umer 

Guest Editor

PLOS ONE